# Alpha-Phellandrene and Alpha-Phellandrene-Rich Essential Oils: A Systematic Review of Biological Activities, Pharmaceutical and Food Applications

**DOI:** 10.3390/life12101602

**Published:** 2022-10-14

**Authors:** Matteo Radice, Andrea Durofil, Raissa Buzzi, Erika Baldini, Amaury Pérez Martínez, Laura Scalvenzi, Stefano Manfredini

**Affiliations:** 1Faculty of Earth Sciences, Universidad Estatal Amazónica, Puyo 160150, Ecuador; 2Department of Life Sciences and Biotechnology, University of Ferrara, 44121 Ferrara, Italy

**Keywords:** alpha-phellandrene, antimicrobial, antitumoral, insecticidal, food-borne pathogens

## Abstract

Alpha-phellandrene is a very common cyclic monoterpene found in several EOs, which shows extensive biological activities. Therefore, the main focus of the present systematic review was to provide a comprehensive and critical analysis of the state of the art regarding its biological activities and pharmaceutical and food applications. In addition, the study identified essential oils rich in alpha-phellandrene and summarized their main biological activities as a preliminary screening to encourage subsequent studies on their single components. With this review, we selected and critically analyzed 99 papers, using the following bibliographic databases: PubMed, SciELO, Wiley and WOS, on 8 July 2022. Data were independently extracted by four authors of this work, selecting those studies which reported the keyword “alpha-phellandrene” in the title and/or the abstract, and avoiding those in which there was not a clear correlation between the molecule and its biological activities and/or a specific concentration from its source. Duplication data were removed in the final article. Many essential oils have significant amounts of alpha-phellandrene, and the species *Anethum graveolens* and *Foeniculum vulgare* are frequently cited. Some studies on the above-mentioned species show high alpha-phellandrene amounts up to 82.1%. There were 12 studies on alpha-phellandrene as a pure molecule showed promising biological functions, including antitumoral, antinociceptive, larvicidal and insecticidal activities. There were 87 research works on EOs rich in alpha-phellandrene, which were summarized with a focus on additional data concerning potential biological activities. We believe this data is a useful starting point to start new research on the pure molecule, and, in particular, to distinguish between the synergistic effects of the different components of the OEs and those due to alpha-phellandrene itself. Toxicological data are still lacking, requiring further investigation on the threshold values to distinguish the boundary between beneficial and toxic effects, i.e., mutagenic, carcinogenic and allergenic. All these findings offer inspiration for potential applications of alpha-phellandrene as a new biopesticide, antimicrobial and antitumoral agent. In particular, we believe our work is of interest as a starting point for further studies on the food application of alpha-phellandrene.

## 1. Introduction

Plants are one of the most important sources of natural compounds used in the discovery of new pharmaceutical agents, cosmetics, and pesticides. Their essential oils (EOs) consist of thousands of components that are potential agents in the treatment of many diseases, such as 1,8-cineole for respiratory diseases, cancer, digestive disorders, dysphoria, AD, and cardiovascular illness, as well as antimicrobial applications [1], or that can be used to produce toxic effects and repellence in insects, such as eugenol [2]. Moreover, geraniol, linalool, citronellol, citronellal and citral represent five of the most important compounds in the perfume industry [3]. Of all these compounds, terpenes are the most prominent, followed by phenylpropanoids. Monoterpenes are the most emblematic in the terpenes group, being widely distributed and sometimes representing 90% of the total oil content. Monoterpenes are known for possessing different properties, such as antioxidant, enzyme inhibitory and antifungal activities, being hepatoprotective, having sedative functions and stimulating glucose uptake and lipolysis [4,5]. Monoterpenes can be divided into regular and irregular monoterpenes and iridoids. Regular monoterpenes are the most common and of these, monoterpene hydrocarbons comprise several important compounds, such as pinene, limonene and phellandrene [6].

Alpha-phellandrene, isolated for the first time from *Eucalyptus phellandra* (now called *E. radiata*) and from which it was named, is one of a pair of phellandrene cyclic monoterpene and double-bond isomers. It can be extracted from the EOs of the turmeric leaf (54%), *Boswellia sacra* (42%), *Eucalyptus elata* (35%), dill weed (30%) and *Eucalyptus dives* (17%) [7,8], but is also found in several different plant species, such as *Cryptomeria japonica, Heracleum antasiaticum, E. camaldulensis, Gossypium hisutum, Cistus ladanifer*, and *Cannabis sativa*, among others [9].

Alpha-phellandrene has been in public use since the 1940s and has been considered a Generally Recognized as Save (GRAS) product from the Federal Emergency Management Agency (FEMA) since 1965. It has recently been reassessed as not being hazardous for public health [10]. It is approved by the FDA for food use (21 CFR 172.515). It is used to produce fragrances, soaps, detergents, creams and lotions, and its biological activity is of interest in agriculture, food and feed applications and its pharmacological properties are of interest to the pharmaceutical and cosmetic industries.

This review collected all the information related to alpha-phellandrene and alpha-phellandrene-rich EOs up until the present, starting from their principal sources, and reviewing their potential pharmaceutical and food applications, focusing on their bioactivities.

## 2. Materials and Methods

The present systematic review was developed according to the PRISMA guidelines [11], selecting articles from the following scientific databases: PubMed (https://pubmed.ncbi.nlm.nih.gov/, accessed on 4–8 July 2022), SciELO (https://scielo.org/, accessed on 4–8 July 2022), Wiley (https://onlinelibrary.wiley.com/, accessed on 4–8 July 2022), WOS (https://www.recursoscientificos.fecyt.es/, accessed on 4–8 July 2022). In order to manage all the bibliographic references, Mendeley software was used and the search for, and selection of, the articles were independently performed by four researchers, avoiding duplicate data. The following keywords were searched individually and in combination: “alpha-Phellandrene”, “biological activity”, “pharmaceutical” and “food”, and the authors decided to consider the literature published in English over the past 20 years, including some older relevant key data. Due to the large number of articles which reported the word “alpha-Phellandrene”, a preliminary selection was performed by selecting the searches which reported the topic alpha-phellandrene in the title and/or the abstract, avoiding articles which did not directly include the molecule in the study, but simply reported it in the discussion section. Articles were grouped according to the following: the source of the molecule; the antimicrobial, antitumoral, biopesticide and repellence, food preservative and other specific activities; and their toxicity. The process of datamining is synthetized in the following flowchart (Figure 1). Tables and paragraphs were prepared to represent the following criteria: the country where the research was performed, EOs with alpha-phellandrene among the first three components, and results concerning biological activity and toxicity. As reported in Figure 1, the above-mentioned criteria allowed the selection of 99 eligible articles, and the exclusion of 206 articles that did not meet the selection methodology, either due to incomplete information or because they simply mentioned data concerning alpha-phellandrene without focusing on the topic of the present study.

## 3. Results

### 3.1. Source

The results from Table 1 show that the most cited species in relation with the presence of alpha-phellandrene in the extracted EO was *Anethum graveolens*, with individuals from Spain and Tasmania showing concentrations of 70.2% and 49.1%, respectively. When looking at the species with the highest percentages, *Foeniculum vulgare* was the one standing above the others with 82.1% of alpha-phellandrene extracted by steam distillation (SD) from the aerial parts of individual plants found in Italy, followed by the above cited *A. graveolens*, 70.2%, *Canarium ovatum* with 64.9% obtained from resin and *Monodora myristica*, with 53% distilled from the seeds. Focusing on the area from which the plants were collected, India was the first country, with 8 studies citing this molecule. Whilst focusing on the regions of the world, as shown in Figure 2, Europe had the highest amount, 11, of studies.

### 3.2. Chemistry and Biosynthesis

Alpha-phellandrene, chemically 5-isopropyl-2-methyl-1,3-cyclohexadiene, is a colourless to slightly yellow mobile liquid with a peppery, woody and herbaceous aroma. Is one of a pair of phellandrene cyclic monoterpene double-bond isomers in which both double bonds are endocyclic, compared to beta-phellandrene, where one of the bonds is exocyclic. Its biosynthesis starts with the transformation of the three highly reactive diphosphates, geranyl diphosphate (GPP), neryl diphosphate (NPP) and linalyl diphosphate (LPP). Preferring LPP as a substrate, a cyclase closes the linear structure to form the menthyl (or alpha-terpinyl) cation which undergoes a 1,3-hydride shift to furnish the phellandryl cation. This intermediate then liberates a proton to obtain the isomeric monocyclic terpene alpha-phellandrene [56]. The molecular structure of alpha phellandrene is shown in Figure 3.

### 3.3. Alpha-Phellandrene Properties as Pure Molecule

#### 3.3.1. Antitumoral Effects

Researching other treatment approaches for cancer, Lin et al. [57] demonstrated that 25.0 mg/kg of alpha-phellandrene administrated orally promoted immune response, enhancing phagocytosis and NK cell activity through increasing levels of T-cells, monocytes and macrophages in BALB/c mice in vivo. Another study, led by Mr. Lin, focused on this property of alpha-phellandrene and its role in gene expression. It was demonstrated that the oral administration of alpha-phellandrene induced immune responses in mice injected with mice WEHI-3 leukemia cells [58], and cycle arrest and apoptosis in the same line of cells in vitro [59]. They also provided important possible molecular mechanisms of how alpha-phellandrene affected gene expression and the possible signaling pathways in this line of tumor cells [60,61]. Finally, a recent study [62] demonstrated that alpha-phellandrene is not only antitumoral, but also exhibits antinociceptive effects. This molecule showed in vitro cytotoxic action on melanoma B-16/F-10 and Sarcoma 180, inhibition of in vivo tumor growth and decreased direct and indirect mechanical nociception in Sarcoma 180-bearing mice with early and advanced tumors, under acute or subacute conditions of treatment, especially at doses of 25.0 and 50.0 mg/kg.

#### 3.3.2. Biopesticide and Repellent Activity

Few are the studies trying to prove the potential of this single molecule as a biopesticide. For instance Plata et al. [63] found promising results against a common crop pest, *Sitophilus granarius* (Granary weevil), where 10.5 µL/mL of pure terpenoid induced the mortality of more than 90% of individuals after 24 h exposure. Similarly, in one of the earliest studies related to the toxicity of this compound, Jung et al. [64] proved that alpha-phellandrene had a lethal concentration 50 (LC_50_) of 0.28 mg/cm^2^ after 24 h contact exposure against adult females of *Blatella germanica* (German cockroach), one of the most common indoor environmental pests. Furthermore, Ali et al. [65] tested the effects of the single molecule, finding a LC_90_ of 19.3 ppm against *Aedes aegypti* and a LC_90_ of 36.4 ppm against *Anopheles quadrimaculatus*. 

A study by Bleeker et al. [66] confirmed the repellent activity of this monoterpene against *Bermisia tabaci* (whitefly), which is responsible for severe losses in crop production and horticultural practices. Another study [67] focused on the repellency to *Stomoxys calcitrans* using alpha-phellandrene alone or in synergy with *Calophyllum inophyllum* nut oil fatty acids or their esters. Bleeker et al. found that the protection time expressed in hours for alpha-phellandrene alone was about 0.5 h in a concentration of 0.25 mg/cm^2^. The same concentration, together with 0.5 mg/cm^2^ of oleic acid, repelled the female flies in 1.9 h, in 1.9 h with 0.5 mg/cm^2^ of linoleic acid, in 1.6 h with the same amount of methyl oleate and in 1.6 h in synergy with 0.5 mg/cm^2^ of methyl linoleate.

#### 3.3.3. Food Preservative

In an attempt of apply this molecule in food preservation, an interesting activity alpha-phellandrene was discovered by a study group in Taiwan [68]. It was found that a higher concentration of 4.0 µg of alpha-phellandrene per gram of white shrimp (*Litopenaeus vannamei*) enhanced their immune response and resistance against the pathogen *Vibrio alginolyticus.*

#### 3.3.4. Supplementary Data Concerning Specific Biological Activities

Some specific activities of this molecule have been tested and are worth mentioning. For instance, a study from Brazil [69] focused on alpha-phellandrene antinociceptive effects considering different mechanisms of actions through the glutamatergic, opioid, nitrergic, cholinergic and adrenergic systems with promising results. Finally, Marrelli et al. [70] proved that alpha-phellandrene has high potential in protecting proteins from heat-induced degradation with an inhibition concentration 50 (IC_50_) of 73.2 µg/mL.

### 3.4. EOs Rich in Alpha-Phellandrene and Their Main Biological Activities

In this section, a dataset of alpha-phellandrene-rich EOs was selected and summarized. Although the EOs mentioned below present various bioactivity data that are not proven as directly attributable to alpha-phellandrene, these preliminary results are a first step towards hypothesizing and proposing new similar studies on the pure molecule. Therefore, this section covers a set of preliminary data suitable for identifying a specific line of research for further studies on the single molecule. Repeating the single-molecule studies by applying techniques such as High Performance Thin Layer Chromatography (HPTLC) bioautographic assay would make it possible to identify whether the biological activity of the preliminary EO studies is attributable in toto, or at least partially, to alpha-phellandrene, or is totally attributable to synergistic effects of the various components. Hence, the following section refers to biological activity studies that could be seen as encouraging new single-molecule studies.

#### 3.4.1. Antimicrobial Activity

As a preliminary result suitable for further investigations, EOs rich in alpha-phellandrene have been shown to have potential properties as antimicrobial agents and food preservatives, and there are several species within the ethnopharmacological knowledge that possess this molecule as one of their principal components. For instance, Gonzales et al. [71] focused on the antimicrobial activity of *Schinus molle* EO where alpha-phellandrene, together with limonene, were the main components with high activity against *Bacillus cereus*, a typical bacterium which produce toxins in gastrointestinal illnesses. Ozcan et al. [53] found that the EO extracted from fennel, which had alpha-phellandrene as a main component, in a concentration of 40 ppm, but ineffective at 10 ppm, completely inhibited mycelial growth of *Alternaria alternata*, a well-known fungus causing leaf spot and other diseases in many different species of plants. It also had different degrees of fungistatic activity depending on the doses. Another species where alpha-phellandrene is the main compound is *Cymbopogon jwarancusa* and Deka Bhuyan and his study group [72] found high antifungal activity in its EO, specifically against two rice pathogens, *Rhizoctonia solani* and *Bipolaris oryzae*, with a minimum inhibitory concentration (MIC) lower than 100 µL/mL. Another common problem in the control of agricultural crop diseases is the production of aflatoxins from fungi, mainly from *Aspergillus flavus* species. Sindhu et al. [31] proved the potential of the EO from turmeric leaves (containing24.4% of alpha-phellandrene) as a bio-preservative for storage of species, such as maize, peanuts ant tree nuts, among others, showing 95.3% and 100% inhibition of aflatoxins at 1.0 and 1.5%, respectively. Another fungal growth inhibition activity of this molecule was tested by Moro et al. [73] using the EO extracted from the fruits of *A. graveolens* against *Penicillium verrucosum*. The results obtained showed a concentration lower than 0.02 µL/mL inhibited 50% of fungal growth. Do Rosario et al. [74] studied the activity of leaf EO from *S. mole*, with a concentration of alpha-phellandrene of 25.9%, against Gram+ and Gram- pathogenic bacteria and food spoilage fungi. The best results were obtained against *Staphylococcus epidermidis* with an 11.3 mm inhibition zone, compared to 17.0 mm in the positive control, Amoxicillin. Regarding its use as a food preservative, the EO displayed strong activities against *Aspergillus niger* and *Rhizopus stolonifera*, for instance. An extended study by Elshafie et al. [51] focused on three different EOs (with alpha-phellandrene within the main components) from the *Schinus* genus in the control of severe phytopathogens and human pathogens. The results showed the best potential use against the fungal mycelial growth of *Colletotrichum acutatum* in a concentration from 250 to 1000 ppm, and were effective against most of the bacteria tested, showing better results on Gram+ than Gram- bacteria. Another study with optimal results concerning the antimicrobial activities of an EO containing alpha-phellandrene was the one from Ennigrou et al. [75], which found optimal activities against the Gram+ bacteria *Enterococcus feacium* and *Streptococcus agalactiae*. Another study, [76], evaluated the antimicrobial activity of another species from the same genus, *S. terebinthifolius*, where the EO composition of the ripe fruit presented a 12.6% concentration of alpha-phellandrene. Cole et al. focused on wild strains of hospital origin bacteria, finding strong activity of the oil against *Corynebacterium* sp. and *Pseudomonas* sp., with an MIC of 3.55 µg/mL and 7.11 µg/mL, respectively. Salem et al. [77] also focused their study on this species where their oil, extracted from ripe fruits, presented 32.8% of alpha-phellandrene and showed am MIC of 16.0 µg/mL and 32.0 µg/mL against *S. aureus* and *P. aeruiginosa*, respectively. Likewise, Celaya et al. [78] studied the effects of the EO from *S. areira* against *S. aureus*. This research studied the antimicrobial properties of different varieties of the plant finding that the leaf EO which contained the highest amount of alpha-phellandrene (30.9%) had optimal MIC against *S. aureus*, but was not the best when compared with varieties also containing higher concentrations of limonene, suggesting that the synergistic effect of these two molecules could be even stronger than the separate effects.

Chaftar et al., [79], compared six different EOs in their activity against *Legionella pneumophila*. They found that the EO from *Juniperus phoenicea*, containing the highest concentration of alpha-phellandrene, presented the lowest MIC against the different pathogen strains tested (MIC < 30.0 µg/mL). Another study, [80], proved the potential of this molecule used both as an antimicrobial and as a food preservative. Eissen et al. tested the EO of *Alchornea cordifolia*, in which alpha-phellandrene was one of the major components, finding strong activity against *S. aureus* (MIC = 78.0 µg/mL) and *A. niger* (MIC = 156.0 µg/mL). Another EO studied for its activity against *S. aureus* was extracted from *Tussilago farfara* (alpha-phellandrene 26%) and showed an MIC_50_ of 368.0 µg/mL. This group, headed by Boucher [81], also studied its activity against *E. coli* with a MIC_50_ of 468.0 µg/mL. Mercier et al. [21] focused, likewise, on these two specific and common pathogens, finding that the EO from *Canarium ovatum*, with 64.9% of alpha-phellandrene, had an MIC_90_ of 180.0 and 310.0 µg/mL against *E. coli* and *S. aureus*, respectively. Further proof of the antimicrobial potential of this molecule was given by Dalli et al. [82], testing *Nigella sativa* seed EO against different bacterial strains, and obtaining promising results against *Acinetobacter baumannii*. Irahal et al. [41] tested the EO extracted from *S. molle* and it showed an MIC against *Staphylococcus aureus* of 0.25 mg/mL. Finally, Lisboa et al. [83] found that the EO from *Rosmarinus officinalis* (alpha-phellandrene 11.1%) could be considered as a potential treatment against endometritis-causing microorganisms in mares.

Singh et al. [84] evaluated the antifungal effects of the EOs extracted from *Senecio amplexicaulis*, which had alpha-phellandrene as its main compound (48.6%). The experiment showed that the EO had strong antifungal activity in a concentration that ranged between 157.0 to 199.3 µg/mL against different fungal species (*Macrophomina phaseolina*, *Rhizoctonia solani*, *Sclerotium rolfsii*, *Fusarium oxysporum* and *Pythium debaryanum*). Likewise, Gakuubi et al. [85] tested the EO from *E. camaldulensis*, with an alpha-phellandrene concentration of 10%, on different species from the *Fusarium* genus showed that the EO, at a concentration of 7.0–8.0 µL/mL after five days of incubation, completely inhibited mycelial growth in all the tested pathogens, and had an MIC in the same range of concentration. Another antifungal activity study of this molecule was led by Cabral et al. [86], focusing on the EO extracted from *Foeniculum vulgare* which displayed its effects against *Cryptococcus neoformans* and *C. albicans* (MIC of 0.32 and 0.64 µL/mL, respectively) and inhibited the filamentation of *C. albicans* with a concentration of 0.08 µL/mL.

#### 3.4.2. Antitumoral Effects

Another important investigation area which involved EOs rich in alpha-phellandrene referred to their antitumoral activities. Bendaoud et al. [50] studied this property from the EOs of two species that have high concentrations of this molecule, *S. molle* and *S. terebinthifolius* (46.5% and 34.4%, respectively). The IC of the EO to stop MCF-7 human breast cancer cell growth was tested. Using *S. terebinthifolius* EO, the IC_50_ was 47.0 µg/mL, whilst it was 54.0 µg/mL when using *S. molle*. Bakarnga-Via et al. [87] focused on the effects of *Monodora myristica* EO on the same line of tumor cells (MCF-7), finding an IC_50_ of 0.265 µL/mL when alpha-phellandrene was at a concentration of 67.1%. Likewise, Hsieh et al. [88] found that this molecule, at a concentration of 30.0 µM, induced necrosis of human liver tumor (J5) cells, increasing ATP depletion via enhanced nitric oxide and reactive oxygen species production and lactate dehydrogenases leakage. In another study [89], the same group suggested that this molecule could induce J5 cell autophagy by regulating mTOR and LC-3II expression, p53 signaling and NF-*κ*B in J5 cells. De Lima et al. [90] studied the effect of the EO from the leaves of *Conobea scoparioides* (alpha-phellandrene 13.1%) against three different lines of human tumor cells, and, in particular, line MCF-7 for human breast adenocarcinoma, line HepG2 for human hepatocellular carcinoma, and HCT116 for human colon carcinoma, finding that the EO displayed in vitro cytotoxicity against all of them.

A study from Jordan [91] found that the EO from *S. molle*, which ranged in concentration of alpha-phellandrene between 40.8% and 55.9%, haf a moderate antiproliferative activity on the colorectal and breast cancer cell lines (HCT116, Caco-2 and MCF7, T47D, respectively) with an IC_50_ ranging between 21.0–65.0 µg/mL. 

#### 3.4.3. Biopesticides and Repellent Activity

The use of natural pesticides based on EOs represents an increasingly suitable method for crop protection and, in this way, EOs rich in alpha-phellandrene have shown promising results. Kiran et al. [92] determined the fumigant toxicity of the EO from *Boswellia carterii*(alpha-phellandrene 6.8%) against legume beetles. They found that 0.096 µL of the EO per ml of air was effective against *Callusbruchus chinensis* after 24 h of treatment, obtaining more than 90% mortality. *B. carterii* EO was, likewise, effective against *C. maculatus*, as the LC_90_ of individuals was even lower (0.075 µL/mL air). Alpha-phellandrene was found to be the main component (28.5%) of the EO obtained from *S. areira* leaves and displayed the lowest concentration to cause the mortality of half of the *Metopolophium dirhadum* nymphs tested, a well-known cereal crop pest, when compared to others EOs. Furthermore, the exact LC_50_ for winged adults was 7.5 mg/mL at 24 h after contact exposure [93]. 

A recent study by Kostic et al. [94] described the potential of dill (*A. graveolens*) seed EO, containing 13.1% alpha-phellandrene, in controlling the spread of gypsy moths (*Lymantria dispar*). Kostic et al. found that dill seed EO was the most effective, compared with other Eos, when introduced as 1% of the larvae diet, resulting in a reduction of 78% of larval molting at a concentration of 0.5%. The EO obtained from *Curcuma longa* leaves, with an alpha-phellandrene concentration of 42%, showed an LC_90_ for *Lucilia cuprina* larvae at 1.5 µL/cm^2^ after 6h exposure [95]. Osanloo et al. [96] formulated a nanoemulsion of dill EO, with an alpha-phellandrene concentration of 20.8%, obtaining an LC_90_ of 65.0 ppm against thirrd and fourth instar larvae of *A. stephensi.* Another study, [97], proved the effectiveness of using an EO made from the seeds of *Nigella sativa* (alpha-phellandrene 14.9%), against the same organism, obtaining an LC_90_ of 172.6 ppm after 24 h exposure. Raj et al. also studied the same effect against *Ae. aegypti* (LC_90_ = 300.8 ppm) and *Culex quinquefasciatus* (LC_90_ = 364.0 ppm). Likewise, Seo et al. [98] focused on the larvicidal effect against the Culicidae family, specifically in the inhibition of the acetylcholine esterase of the Asian tiger mosquito (*Ae. albopictus*). This group found that dill EO with alpha-phellandrene as the main component, had strong larvicidal activity (>90% mortality) at a concentration of 0.1 mg/mL. To prevent *Ae. aegypti* from oviposition, Santos da Silva et al. [99] used the EO from *Commiphora leptophloeos* leaves, with 26.3% of alpha-phellandrene, and found strong oviposition deterrent effects in a concentration between 25.0 and 100.0 ppm. Their bioassays also showed good larvicidal activity with an LC_50_ of 99.4 ppm. Against the larvae of mosquitos, the EO from the leaves of *Echinophora lamondiana* (alpha-phellandrene 14.1%) showed an LC_90_ of 169.1 and 46.0 ppm against *Ae. Aegypti* and *A. quadrimaculatus*, respectively. Furthermore, Do Nascimento et al. [100] studied the larvicidal effects of the EO from *Piper klotzschianum* seeds with a concentration of alpha-phellandrene of 17% against *Ae. aegypti*. It was found that 13.3 µL/mL was able to kill 50% of individuals after 48h incubation with the EO. This group also studied the lethality against *Artemia salina*, which is usually known as ‘the brine shrimp lethality assay’, where the LC_50_ was 14.0 µL/mL. 

Alpha-phellandrene has also been tested as a repellent against various organisms. For instance, the ethyl acetate extracts from *Myrica gale* inflorescences (alpha-phellandrene 25%) repelled 82.1% of mosquitos (*Ae. aegypti*) tested in a study lead by Jaenson et al. [101]. Ashitani et al. [102] proved the repellency activity of the EO extracted from the fresh leaves of *Hyptis suaveolens* against *Ixodes ricinus*, where the concentration of alpha-phellandrene was about 28.3%, determining that the EO was effective against 81% of the individuals tested.

#### 3.4.4. Food Preservative

Another potential application of EOs rich in alpha-phellandrene is their use as natural preservative ingredients in food. For instance, a study from Tunisia, [49], found optimal results focusing on the properties of the EO from *S. molle*, with a concentration of alpha-phellandrene of 35.9%, in the inhibition of *Salmonella* genus growth in raw beef. In another direction of food preservation, Olomedo et al. [103] proved that the EO from *S. molle* with alpha-phellandrene within the main components improved the stability of fried salted peanuts by preventing lipid oxidation and development of rancid flavors. In a recent study, Teneva et al. [20], focused on biological preservation of mayonnaise and found interesting results with a combination of the probiotic *Lactobacillus plantarum*, basil, and dill EOs (22.7% alpha-phellandrene in the latter) significantly decreasing the number of undesired microflora, making this combination an effective biological preservative for mayonnaise. Benkhoud et al. [104] incorporated thyme EO (1.3% alpha-phellandrene) to extra virgin olive oil proving its ability to preserves its nutritional quality, and sensory attributes, and, most of all, to improve its oxidative stability during long-term storage. Tavakkoli et al. [34] evaluated the effects of tomato residuum extract (TRE) dipping and Arabic gum (AG) coating enriched with dill EO (DEO), with alpha-phellandrene as a main component, on the shelf-life extension of refrigerated trout fillets, and found that treatments TRE 3%-AG-DEO 2% and TRE 6%-AG-DEO 2% had significantly effects.

#### 3.4.5. Miscellaneous Data Concerning Specific Biological Activities

There are some particular activities of EOs rich in alpha-phellandrene worth mentioning as potential preliminary data. For instance, El Ayeh-Zakhama et al. [48] studied the allelopathic effects of an EO from *Citharexylum spinosum* where alpha-phellandrene was one of the main compounds, and discovered that it inhibited the early growth of lettuce seedings. Specifically, 0.4 mg/mL of flower EO induced a percentage inhibition of root length of 72.4%. Hajhashemi and Abbasi [35] found that *A. graveolens* EO, with alpha-phellandrene as a main compound, significantly reduced, in a dose-dependent manner, total cholesterol and triglyceride of rats on a hyperlipidemic diet. Ayelo et al. [105] proved that a blend of four components having alpha-phellandrene, there was high potential for use as a kairomone-based lure to recruit *Nesidiocoris tenuis*, the predator of tomato pests (*Tuta absoluta* and *Trialeurodes vaporarioum*) in an attempt at biological control to reduce crop damage.

### 3.5. Toxicity

Besides all the possible biological activities of this molecule, there is also evidence of its mutagenic and carcinogenic toxicity, for instance after metabolic activation. Mademtzoglou et al. [106] found evidence of alpha-phellandrene genotoxicity through the *Drosophila* wing spot test, finding that the frequency of mutant spots significantly increased, when compared with a negative control, even at the lowest concentration (1.5 µL/mL). Another study, [107], investigated the contact allergy produced by the topical use of EOs and found that alpha-phellandrene is one of the most frequently reacting sensitizers, with a range of 31 to 63% of positive reactions in all the patients tested. These results are very relevant to the development of new products derived from alpha-phellandrene or alpha-phellandrene-rich essential oils. Consequently, researchers are encouraged in investigating, with exhaustive toxicological evaluations, long-term and acute toxicity. 

## 4. Discussion

The aims of this review, were to attract interest in alpha-phellandrene and, in a wider context, to highlight its potential applications in the pharmaceutical and food industries. There are preliminary data concerning the potential of alpha -Phellandrene, as a pure compound, in antitumoral, antinociceptive, larvicidal and insecticidal applications. Additionally, several EOs, in which alpha-phellandrene is one of the main compounds, show promising results and represent a preliminary dataset to identify new active molecules.

According to ISO 9235:2021, EOs must be described by the following extraction methods: steam distillation, dry distillation and mechanical extraction from the epicarp of citrus fruits. The present definition is useful in order to distinguish an EO from similar vegetal extracts, such as absolutes, alcoholates, concretes and oleoresins [108]. EOs are volatile and oily liquids easily distinguishable due a typical fragrance and can be obtained from a large number of plants and different anatomical parts, such as seeds, leaves, flowers, barks, fruits and roots. As reported by Sadgrove and Jones [109], at the beginning of the 16th century the concept of EOs was conceived by a Swiss scientific pioneer studying a drug called “Quinta essentia”. Concerning the chemical composition, EOs are complex mixtures of several volatile compounds, including monoterpenes, sesquiterpenes, esters, ketones, aldehydes and alcohols. Among the many EO activities, the antimicrobial activity is very interesting for its possible applications to humans, animals, and food preservation and shows relevant variability, due to the heterogeneous chemical composition of the EOs, the chemotypes, and the different conditions relating to the cultivation, harvesting and processing of plant material. As reported by several authors [110,111], EO antimicrobial activity is probably linked to damage to the cytoplasmic membrane, interference with ion exchange and others enzyme inhibitory effects and modifications in ATP levels. The aforementioned mechanisms of action interfere with the permeability of the cell membrane to such an extent that the microorganism dies. In the present review article, alpha-phellandrene has been frequently mentioned as one of the main compounds of EOs with antimicrobial activity, showing, in some cases, a broad spectrum of activity and synergistic effects with other terpenes. Although the data must be complemented by antimicrobial activity studies on the pure molecule or by a High-Performance Thin Layer Chromatography (HPTLC) bioautographic assay [112,113,114], the preliminary activities collected in this study represent, in our opinion, an interesting starting point encouraging further research.

Antitumoral activities have also been frequently mentioned as a promising field of study, both for pure molecules and EOs rich in alpha-phellandrene. As reported by Lesfgards et al. [115], several searches have focused the in vivo anticancer activity of terpenoids contained in EOs, including some promising clinical studies on humans. The mode of action seems to be related to the activation of proapoptotic processes in the mitochondria in cancer cells, inhibition of tumor-inducing genes and production of free radicals in cancer cells. Additionally, EOs present minor undesired effects on healthy cells, probably due to a regulation of liver detoxification enzymes. Another study, performed by Spinsi et al. [116], investigated the preventive and therapeutic activity of various EOs and their components against colorectal cancer (CRC), showing interesting preliminary results with carvacrol, geraniol, cinnamaldehyde and beta-caryophyllene. Taken together, these results suggest new therapeutic approaches, which involve EOs or their individual compounds, in CRC prevention and treatment, and the possibility to explore decreasing the doses of common chemotherapy drugs with the related side effects. EOs and their constituents are recognized as antioxidant agents, playing a key role in preventing chronic intestinal oxidative damage and chronic low-grade inflammation, two of the main risk factors in CRC. Similar conclusions have also been put forward in the treatment of melanoma, where, again, the antioxidant activity of EOs is identified as a determining factor in reducing melanogenesis. Overproduction of melanin induces hyperpigmentation and promotes the development of melanoma. Other studies indicate that EOs can reduce angiogenesis, lymphangiogenesis and tumor metastasis proliferation [117]. With regard to the mechanisms of action of EOs and their individual compounds, although it is rather difficult to understand which molecule may exert anticancer activity, comforting results were found in the case of the EO of *Origanum vulgare* [118], *Rosmarinus officinalis* [119], *Lippia microphylla* [120] and the *Mentha* and *Thymus* genus [121,122]. On the basis of these findings and the data reported in this study, further research in the antitumoral field would be appropriate for both alpha-phellandrene and EOs particularly rich in this molecule.

Alpha-phellandrene and oils rich in this molecule have proven to be potential biopesticides, larvicides and insect repellents. Furthermore, considering the research conducted against food-borne pathogens, it can be stated that alpha-phellandrene could have promising applications in the food and agro-industrial supply chains, ranging from biological treatment of crops to pre- and post-harvest treatments. The attention of researchers to these new applications of EOs is well documented [123,124,125].

Furthermore, the larvicidal efficacy of alpha-phellandrene opens up new research perspectives in the control of tropical disease vectors and human and animal health. These areas of research are relatively new for EOs and their components, which are traditionally marketed in the cosmetics and perfume sectors, but they represent innovative applications already supported by a growing body of scientific literature [126,127,128,129,130]

Finally, alpha-phellandrene deserves further investigation in relation to its antinociceptive properties and toxicological profile, which would provide a fundamental basis for further pharmaceutical developments.

## 5. Conclusions

Several authors have investigated the alpha-phellandrene’s biological activity and there is an abundant data set of EOs rich in this molecule being studied. This review work has drawn attention to the most promising data, with the aim to suggest new research trends. In particular, further studies should be encouraged concerning the following: (1) the antimicrobial and antitumoral effects of alpha-phellandrene as a pure compound and in synergy with drugs; (2) the development of new biological activity studies on EOs richer in alpha-phellandrene, by applying analytical techniques, such as the High Performance Thin Layer Chromatography (HPTLC) bioautographic methodology; (3) the larvicides and insect repellent activity of alpha-phellandrene potential as plant-based biopesticides; (4) potential application as novel food preservatives and (5) exhaustive toxicological evaluations on long-term and acute toxicity.

## Figures and Tables

**Figure 1 life-12-01602-f001:**
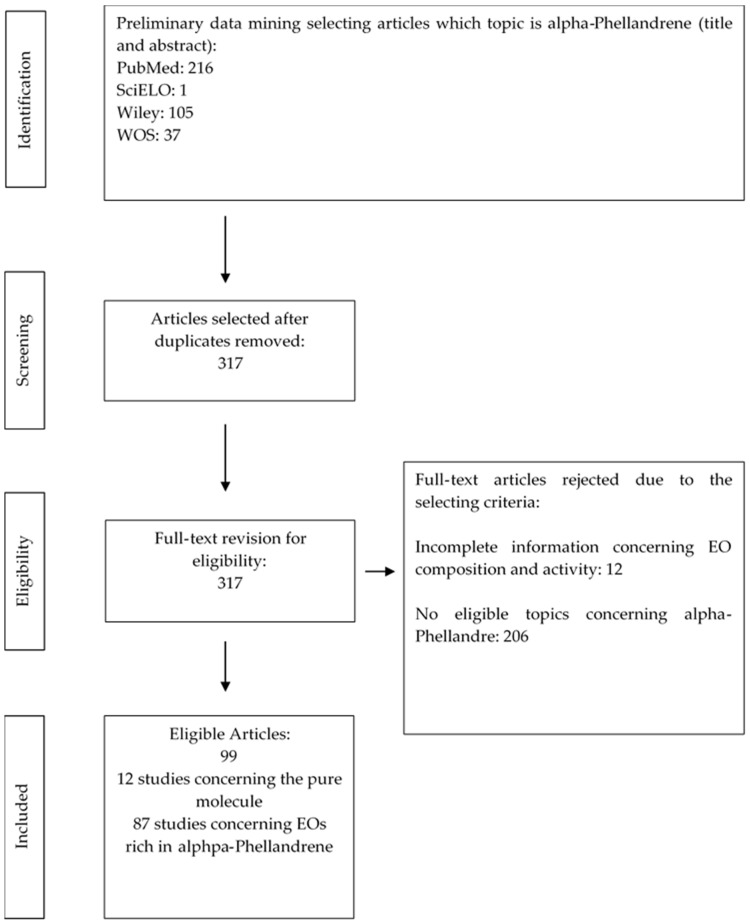
Flowchart for the search and selection of studies considered in the review.

**Figure 2 life-12-01602-f002:**
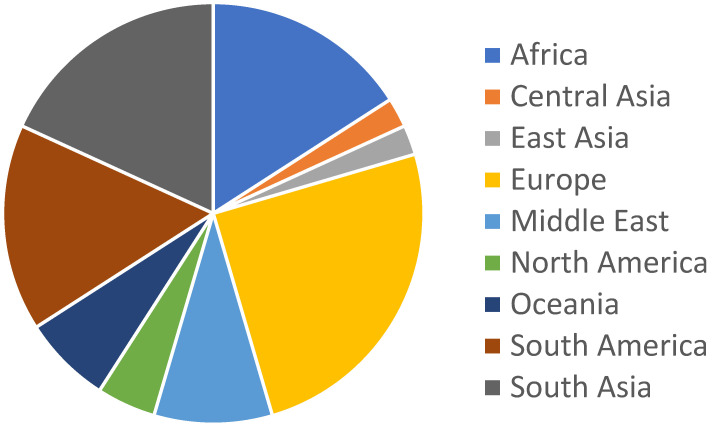
Biogeographic precedence of plant material in the reviewed papers.

**Figure 3 life-12-01602-f003:**
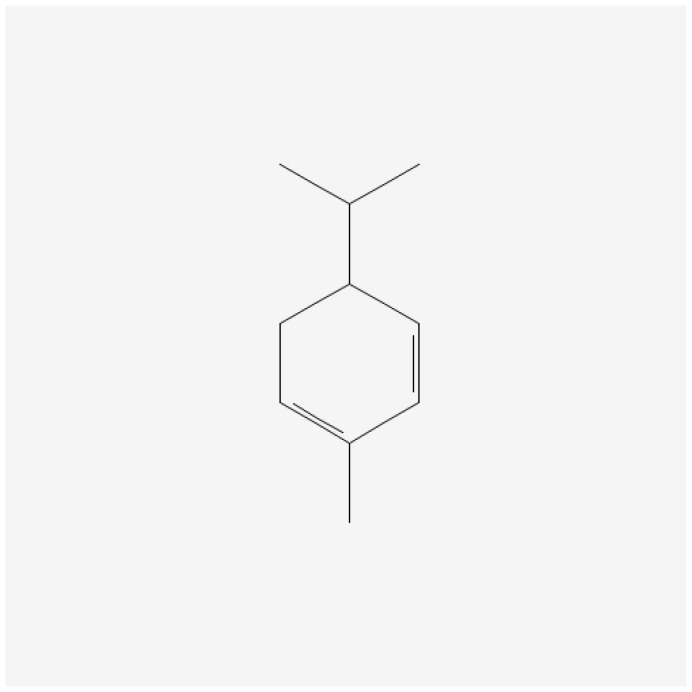
Molecular structure of alpha-phellandrene Source: https://pubchem.ncbi.nlm.nih.gov/compound/7460#section=2D-Structure (accessed on 5 October 2022).

**Table 1 life-12-01602-t001:** EOs reported with alpha-phellandrene among the first three components.

Country	Species	Part	Extract	Alpha-Phellandrene (%)	Ref.
Alabama	*Curcuma longa*	Rhizomes	HD	11.8	[12]
Algeria	*Pituranthos scoparius*	Stems	HD	7.1	[13]
Argentina	*Eucalyptus tereticornis*	Seeds	HD	9.4	[14]
Australia	*Eucalyptus calcicola*	Leaves	VD	11.0	[15]
Australia	*Eucalyptus incerata*	Leaves	VD	6.3	[16]
Brazil	*Aniba rosiodora*	Aerial parts	HD	22.8	[17]
Brazil	*Lantana camara*	Aerial parts	HD	16.4	[18]
Brazil	*Piper diltatum*	Leaves	HD	22.5	[19]
Bulgaria	*Anethum* *graveolens*	Aerial parts	HD	22.7	[20]
Canada	*Canarium ovatum*	Resin	HD	64.9	[21]
Colombia	*Lippia origanoides*	Aerial parts	HD	13.0	[22]
Ecuador	*Gynoxys miniphylla*	Leaves	SD	16.6	[23]
France	*Limbarda crithmoides*	Aerial parts	HD	11.9	[24]
Germany	*Anethum* *graveolens*	Aerial parts	SD	32.3	[25]
India	*Altingia excelsa*	Leaves	HD	15.9	[26]
India	*Angelica glauca*	Aerial parts	HD	13.5	[27]
India	*Anethum* *graveolens*	Aerial parts	HD	19.1	[28]
India	*Callistemon lanceolatus*	Leaves	HD	5.8	[29]
India	*Cinnamomum zeylanicum*	Leaves	HD	6.2	[30]
India	*Curcuma longa*	Leaves	HD	24.4	[31]
India	*Eucalyptus camaldulensis*	Aerial parts	HD	27.5	[32]
India	*Hyptis suaveolens*	Leaves	HD	22.8	[33]
Iran	*Anethum graveolens*	Aerial parts	HD	30.2	[34]
Iran	*Anethum graveolens*	Aerial parts	HD	32.0	[35]
Italy	*Elaeoselinum ascelpium*	Flowers	HD	42.5	[36]
Italy	*Elaeoselinum ascelpium*	Leaves	HD	11.0	[36]
Italy	*Foeniculum vulgare*	Aerial parts	SD	82.1	[37]
Italy	*Foeniculum vulgare*	Stems	HD	36.9	[38]
Italy	*Ridolfia segetum*	Flowers	HD	19.4	[39]
Italy	*Ridolfia segetum*	Stems	HD	12.9	[39]
Japan	*Petasites japonicus*	Flower stems	HD	11.0	[40]
Morocco	*Schinus molle*	Aerial parts	HD	9.6	[41]
Nigeria	*Monodora myristica*	Seeds	n.r.	53.0	[42]
Portugal	*Eucalyptus globulus*	Fruits	SD	17.2	[43]
Portugal	*Lavandula pinnata*	Flowers	SD	15.9	[44]
Spain	*Anethum graveolens*	Aerial parts	HD	70.2	[45]
Spain	*Sideritis hirsute*	Aerial parts	SD	9.2	[46]
Tasmania	*Anethum graveolens*	Aerial parts	n.r.	49.1	[47]
Tunisia	*Citharexylum spinosum*	Roots	HD	30.8	[48]
Tunisia	*Schinus mole*	Fruits	HD	35.9	[49]
Tunisia	*Schinus molle*	Fruits	SD	46.5	[50]
Tunisia	*Schinus mole*	Leaves	SD	35.7	[51]
Tunisia	*Schinus terebinthifolius*	Fruits	SD	34.4	[50]
Tunisia	*Schinus terebintifolius*	Fruits	SD	44.3	[51]
Turkey	*Echinophora tenuifolia*	Aerial parts	SD	51.0	[52]
Turkey	*Foeniculum vulgare*	Flowers	HD	5.8	[53]
Uzbekistan	*Heracleum lehmannianum*	Aerial parts	HD	10.5	[54]
Venezuela	*Espeletia schultzii*	Leaves	HD	52.4	[55]

HD: hydro-distillation; SD: steam-distillation; VD: vacuum-distillation; n.r.: not reported.

## Data Availability

Not applicable.

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
