# Peer review of "Alpha-Phellandrene and Alpha-Phellandrene-Rich Essential Oils: A Systematic Review of Biological Activities, Pharmaceutical and Food Applications"

_life, 2022, doi:10.3390/life12101602_

Round 1

Reviewer 1 Report

I found the article very good. 

Very interesting.

Author Response

Comments/suggestions Reviewer 1 Corrections
I found the article very good.
Very interesting.                                  

Authors’ response: Thank you for the favourable comment

Reviewer 2 Report

in line 39 please replace bacilli with antimicrobial

please insert the appropriate reference for the following paragraph turmeric leaf 50 (54%), Boswellia sacra (42%), Eucalyptus elata (35%), dill weed (30%) and Eucalyptus dives 51 (17%) [7,8] but is also found in several different plant species such as Cryptomeria japonica, 52 Heracleum antasiaticum, Eucalyptus camaldulensis, Gossypium hisutum, Cistus ladanifer, and 53 Cannabis sativa between others

and delete the following sentence in line 54 page 2

 (https://lotus.naturalproducts.net/compound/lo- 54 tus_id/LTS0234318) or (PubChem).

mention the full name of GRAS and FEMA

in line 183 the percentage is missed maybe (% of alpha-phellandrene)

the authors have highlighted the importance of alpha-phellandrene but they didn't show sufficient criticism about the link between its toxicity and the justification of its large use !! what are the future prospective to overcome this problem Toxicity/rational use)

Author Response

Comments/suggestions Reviewer 2                

Corrections
in line 39 please replace bacilli with antimicrobial

AR: Thank you for the comments

Correction made

please insert the appropriate reference for the following paragraph turmeric leaf 50 (54%), Boswellia sacra (42%), Eucalyptus elata (35%), dill weed (30%) and Eucalyptus dives 51 (17%) [7,8] but is also found in several different plant species such as Cryptomeria japonica, 52 Heracleum antasiaticum, Eucalyptus camaldulensis, Gossypium hisutum, Cistus ladanifer, and 53 Cannabis sativa between others

Correction made
and delete the following sentence in line 54 page 2
(https://lotus.naturalproducts.net/compound/lo- 54 tus_id/LTS0234318) or (PubChem)

Correction made
mention the full name of GRAS and FEMA

Correction made
in line 183 the percentage is missed maybe (% of alpha-phellandrene)

Correction made
the authors have highlighted the importance of alpha-phellandrene but they didn't show sufficient criticism about the link between its toxicity and the justification of its large use !! what are the future prospective to overcome this problem Toxicity/rational use)

AR: Thank you for the comments
The point is really important, in fact, in the conclusions, under point 5, it is specified that among the actions to be taken, it is necessary to implement an exhaustive toxicological evaluation on long-term and acute toxicity. Moreover, we highlighted this killer point in the paragraph 5 (line 675).

Reviewer 3 Report

Dear Authors,

I have read the paper entitled “alpha-Phellandrene: a review of source, biological activities, pharmaceutical, cosmetic and food application”.  After careful revision, I found the following inconsistencies which make the paper unsuitable for publication in Life.

Most of the papers included in the review refer to essential oils or extracts which have as constituent alpha-phellandrene. One does not attribute the biological properties of multi-component plant-derived extractives (either essential oil or extract) to only one compound, in this case alpha-phellandrene. In such mixtures we discuss about the synergy between constituents. I strongly advise you to revise the literature included in your review.

I encourage authors to address these points in order to complete the review of alpha-phellandrene properly.

Good luck!

Author Response

Comments/suggestions Reviewer 3 Corrections
Dear Authors,
I have read the paper entitled “alpha-Phellandrene: a review of source, biological activities, pharmaceutical, cosmetic and food application”.  After careful revision, I found the following inconsistencies which make the paper unsuitable for publication in Life.
Most of the papers included in the review refer to essential oils or extracts which have as constituent alpha-phellandrene. One does not attribute the biological properties of multi-component plant-derived extractives (either essential oil or extract) to only one compound, in this case alpha-phellandrene. In such mixtures we discuss about the synergy between constituents. I strongly advise you to revise the literature included in your review.
I encourage authors to address these points in order to complete the review of alpha-phellandrene properly.
Good luck!

AR: Thank you for the comments
We agree that the synergistic effects of essential oils are not directly attributable to a single molecule. Accordingly, we have separated the data on the bioactivities of the pure molecule and interpreted the data on essential oils as preliminary information on alpha-phellandrene-rich oils. These data could serve as a guide for further studies on the pure molecule. All the text, including the Flowchart for the search and selection of studies, have been improved in order to clarify that the present work has selected and identified scientific data concerning the pure molecule and alpha phellandrene-rich oils, correcting and eliminating confusing statements concerning bioactivities in the discussion. Your input was really helpful in improving the text and we believe it is now much more relevant to the intentions of our work

Hoping that the revised version of manuscript life-1924091 will be now eligible for publication in the journal ‘Life’, we convey our best regards.

Sincerely,

Matteo Radice 

Round 2

Reviewer 3 Report

Dear Authors,

the manuscript has been significantly improved, hence is now acceptable for publication in Life.